# Modeling Conceptual Attribute Likeness and Domain Inconsistency for Metaphor Detection

**Yuan Tian[1,2], Nan Xu[1,3]\*, Wenji Mao[1,2]\*, Daniel Dajun Zeng[1,2]**

[1]Institute of Automation, Chinese Academy of Sciences
[2]School of Artificial Intelligence, University of Chinese Academy of Sciences
[3]Beijing Wenge Technology Co., Ltd
{tianyuan2021,xunan2015,wenji.mao,dajun.zeng}@ia.ac.cn

## Abstract

Metaphor detection is an important and challenging task in natural language processing, which aims to distinguish between metaphorical and literal expressions in text. Previous studies mainly leverage the incongruity of source and target domains and contextual clues for detection, neglecting similar attributes shared between source and target concepts in metaphorical expressions. Based on conceptual metaphor theory, these similar attributes are essential to infer implicit meanings conveyed by the metaphor. Under the guidance of conceptual metaphor theory, in this paper, we model the likeness of attribute for the first time and propose a novel **A**ttribute l**I**keness and **D**omain **I**nconsistency **L**earning framework (AIDIL) for word-pair metaphor detection. Specifically, we propose an attribute siamese network to mine similar attributes between source and target concepts. We then devise a domain contrastive learning strategy to learn the semantic inconsistency of concepts in source and target domains. Extensive experiments on four datasets verify that our method significantly outperforms the previous state-of-the-art methods, and demonstrate the generalization ability of our method.

## 1 Introduction

Metaphor is pervasive in various forms of language expressions, such as political speech, advertising text, and literary work. Merriam-Webster Dictionary defines *metaphor* as "a figure of speech in which a word or phrase literally denoting one kind of object or idea is used in place of another to suggest a *likeness* or analogy between them"[1]. Metaphor detection aims to distinguish between metaphorical and literal expressions in text. It has gained increasing research interest in recent years and plays a vital role in various NLP applications that need to understand implicit semantics, such as

---

*Corresponding author
[1]https://www.merriam-webster.com/dictionary/metaphor

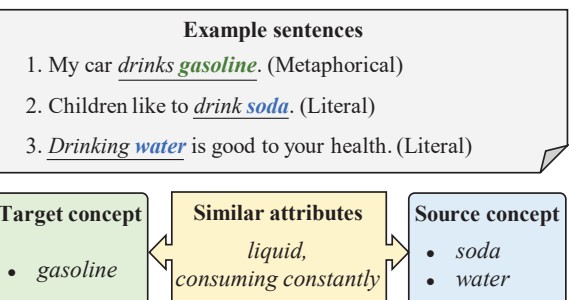

Example sentences

1. My car *drinks **gasoline***. (Metaphorical)
2. Children like to *drink **soda***. (Literal)
3. *Drinking **water*** is good to your health. (Literal)

| Target concept | Similar attributes | Source concept |
|---|---|---|
| • *gasoline* | *liquid, consuming constantly* | • *soda* 
 • *water* |

Figure 1: Illustration of metaphor understanding with similar attributes. The target concept *gasoline* and the source concept (e.g., *soda* and *water*) are domain-inconsistent with their intermediate similar attributes (e.g., *liquid* and *consuming constantly*). These similar attributes help understand the implicit meaning conveyed by the metaphor, that is, drinking gasoline implies consuming gasoline a lot like consuming liquid.

---

machine translation (Mao et al., 2018), sentiment analysis (Mao and Li, 2021), and dialogue system (Sun et al., 2023).

Existing research on metaphor detection mainly focuses on detecting metaphors at word-pair level (Rei et al., 2017; Su et al., 2020; Ge et al., 2022) and token-level (Su et al., 2021; Choi et al., 2021; Zhang and Liu, 2022). Early studies construct linguistic features using manually-created resources for metaphor detection, such as abstractness (Turney et al., 2011), imageability (Tsvetkov et al., 2014) and visibility (Shutova et al., 2016). Recent studies utilize metaphor identification theories (Su et al., 2021; Choi et al., 2021; Zhang and Liu, 2022), including selection preference violation (SPV) (Wilks, 1975, 1978) and metaphor identification procedure (MIP) (Group, 2007), to help design deep learning methods for metaphor detection. Although these methods have achieved promising performances, their underlying theories only utilize contextual clues relevant to the basic meaning of words and their contexts for metaphor identification, lacking the deeper level understanding of

conceptual meanings and semantic characteristics associated with metaphor.

In contrast to the above metaphor identification theories, a more explainable theory, conceptual metaphor theory (CMT) (Lakoff and Johnson, 2008), argues that metaphor facilitates a mapping of a set of similar attributes between the concepts in two different domains (i.e. source and target domains). Inspired by CMT, Ge et al. (2022) generate the source and target concepts from different domains to benefit metaphor detection. However, their work only considers domain-inconsistent information, ignoring the mapping of similar attributes shared by source and target concepts, which provides essential clues to infer the implicit meanings that metaphors tend to convey.

According to CMT, metaphor understanding implies a plausible mapping of a set of similar attributes between the concepts in source and target domains. Hence, *attribute* in CMT is a broad characteristic obtained by activating human imagination and association based on commonsense knowledge as well as perceptual experience. Fig. 1 gives an example to illustrate metaphor understanding with similar attributes. In this example, attribute likeness (*liquid* and *consuming constantly*) is the intermediate association to the implicit meaning (*consuming a lot*) conveyed by this metaphor. Therefore, attribute likeness functions as important associated information for metaphor detection, and the mining of this attribute similarity strongly depends on an appropriate refinement process from plentiful candidate attributes.

Based on the above considerations, in this paper, we propose an **A**ttribute l**I**keness and **D**omain **I**nconsistency **L**earning framework, namely AIDIL, for word-pair metaphor detection. We devise an attribute siamese network with layer-wise attribute refinement graphs to model attribute likeness information and incrementally filter irrelevant information from plentiful candidate attributes. Meanwhile, to exploit information of inconsistent domains, we design a domain contrastive learning strategy on the hidden representations of concepts from source and target domains so as to learn an embedding space with information of domain inconsistency. The two components jointly learn the association and disparity between source and target concepts for effective metaphor detection. Additionally, our method is also capable of providing explainable clues like source concepts or learned attribute information in

metaphor prediction. The main contributions of our work are as follows:

- Inspired by CMT, we identify that metaphor implies not only inconsistent domains but also similar attributes between source and target concepts, and make the first attempt to model attribute likeness for metaphor detection.

- We propose a novel attribute siamese network to incrementally mine similar attributes between source and target concepts via layer-wise attribute refinement graphs on plentiful candidate attributes.

- Extensive experiments verify that our method achieves significant improvements over previous SOTA methods and also demonstrates its generalization ability on unseen data.

## 2   Related Work

Metaphor detection can be roughly divided into two categories: word-pair level and token level. The former determines whether a word pair is metaphorical or literal, and the latter aims to find metaphorical words in sentences. As word pair is a basic type and the most commonly used linguistic form to express metaphor (Tsvetkov et al., 2014), its metaphor detection is the focus of our work.

**Word-Pair Metaphor Detection**   Early studies exploit supervised machine learning methods by constructing word embeddings from linguistic features or external knowledge which are relevant to metaphor, such as embeddings of abstractness (Turney et al., 2011), imageability (Tsvetkov et al., 2014), visibility (Shutova et al., 2016) or property norm (Bulat et al., 2017). After that, some researchers explore indicative clues of metaphor for this task. Based on the observation of cosine similarity between words in a word pair indicating its metaphoricity, Rei et al. (2017) design a neural network with a gating function to model this characteristic. Considering the relation between concreteness and metaphor, Su et al. (2020) construct image representations for concrete word pairs and devise a multimodal model to detect metaphor.

Different from above approaches using intuitive clues for metaphor detection, Ge et al. (2022) adopt a widely accepted metaphor theory—conceptual metaphor theory (CMT)—in their method via generating plausible concepts in source and target domains to help metaphor prediction, which is the

previous state-of-the-art method. However, according to CMT, a metaphor not only indicates domain-inconsistent information but also implies a mapping process of *similar attributes* shared between source and target concepts, which is necessary for explaining the implicit meanings conveyed by the metaphor. All the aforementioned approaches neglect this important information of *similar attributes* for metaphor detection. Thus based on CMT, we focus on mining the attribute likeness between source and target concepts with the disparity of domains for metaphor detection.

**Token-Level Metaphor Detection** Considering special clues related to metaphor, some studies utilize external knowledge resources to benefit this task, such as definitions of words (Su et al., 2021) or multiword expressions (Rohanian et al., 2020). Other studies employ shared features learned from the multi-task framework with the task of word sense disambiguation (Le et al., 2020) or sentiment analysis (Mao and Li, 2021) to detect metaphor. Recent studies adopt metaphor identification theories, including selection preference violation (SPV) (Wilks, 1975, 1978) and metaphor identification procedure (MIP) (Group, 2007), to help design networks (Su et al., 2021; Choi et al., 2021; Zhang and Liu, 2022; Wang et al., 2023; Li et al., 2023). Although the above methods inspired by SPV and MIP have achieved the state-of-the-art results, they identify metaphors based on the basic meaning of the words and their contexts, which are surface-level clues in contrast to the conceptual meanings of attributes and domains in CMT. Therefore, in addition to using these SOTA methods as strong baselines for our experiments, we utilize the conceptual attribute and domain information, and develop the computational method to model attribute likeness and domain inconsistency for effective metaphor detection.

## 3 Problem Definition

In a word pair, we define the verb in a verb-noun pair or the adjective in an adjective-noun pair as the **core word**. According to CMT, the other noun is the **target concept** in a metaphorical word pair or the **source concept** in a literal word pair. Formally, $\mathcal{D}_{tr}=\{(p_k,l_k)\}_{k=1}^{N_{tr}}$ is the training dataset with $N_{tr}$ instances, where $p_k$ is the $k$-th word pair and $l_k$ is the label (metaphorical or literal). Every word pair contains a core word $w_{cor}$ and a concept word $w_{con}$. $\mathcal{D}_{te}=\{(p_k,l_k)\}_{k=1}^{N_{te}}$ is the test dataset with

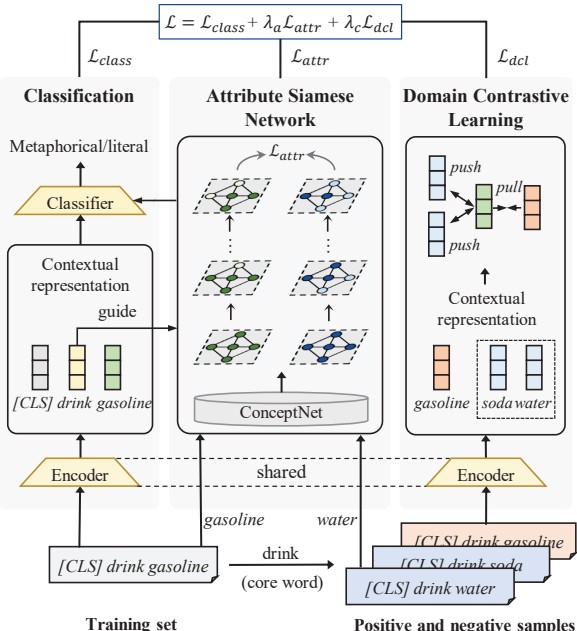

Figure 2: Overall architecture of our proposed AIDIL framework for word-pair metaphor detection.

$N_{te}$ instances. The goal of word-pair metaphor detection is to predict the label of each word pair in $\mathcal{D}_{te}$ by training a model on $\mathcal{D}_{tr}$.

## 4 Method

We propose an **A**ttribute l**I**keness and **D**omain **I**nconsistency **L**earning framework, namely AIDIL, for word-pair metaphor detection. Fig. 2 shows its overall architecture, which contains three primary components: (1) *Attribute siamese network*, which obtains the attribute vector via refining similar attributes and filtering irrelevant attributes gradually based on two attribute graph sub-networks of source concept and target concept respectively guided by the core word; (2) *Domain contrastive learning*, which performs contrastive learning on representations of concepts from source and target domains to learn the information of domain inconsistency for better generalization of contextual representations; and (3) *Classification*, which utilizes contextual representations and the attribute vector to predict the label (metaphorical/literal).

### 4.1 Encoding

To train our model from a good embedding start, we adopt the pre-trained language model BERT (Devlin et al., 2019) as the text encoder. Given a word pair $p$, following Zhong and Chen (2021), we introduce 4 marker tokens ([cor], [/cor], [con] and [/con]) to explicitly mark the boundaries of the core

word $w_{cor}$ and the concept word $w_{con}$ in $p$. The word pair with marker tokens is formulated as

$$\hat{p} = [\text{cor}] \, w_{cor} \, [/\text{cor}] \, [\text{con}] \, w_{con} \, [/\text{con}]. \quad (1)$$

We connect $\hat{p}$, the part-of-speech label of core word $w_{pos}$ (*verb* or *adjective*), the basic definition of core word $d_{cor} = \{w_1^r, w_2^r, \cdots, w_{N_r}^r\}$ with $N_r$ words, the basic definition of concept word $d_{con} = \{w_1^c, w_2^c, \cdots, w_{N_c}^c\}$ with $N_c$ words, and other special tokens as a sequence of input $T$:

$$T = [\text{CLS}] \, \hat{p} \, [\text{SEP}] \, w_{pos} \, [\text{SEP}] \, d_{cor} \, [\text{SEP}] \, d_{con}, \quad (2)$$

where [CLS] and [SEP] are the classification token and separation token respectively. We use the segment embedding of BERT to distinguish among special tokens, core word and its basic definition, concept word and its basic definition, and part-of-speech token. We use WordPiece to split $T$ into tokens and feed them into BERT to obtain contextual representations $H$, as well as contextual representations of core word $\boldsymbol{h}_{cor}$, concept word $\boldsymbol{h}_{con}$, and token [CLS] $\boldsymbol{h}_{[\text{CLS}]}$:

$$H = \text{BERT}(T) = [\boldsymbol{h}_1, \ldots, \boldsymbol{h}_N]^\top \in \mathbb{R}^{N \times d}, \quad (3)$$

$$\boldsymbol{h}_{cor} = \boldsymbol{h}_{[\text{cor}]} + \boldsymbol{h}_{[/\text{cor}]} \in \mathbb{R}^d, \quad (4)$$

$$\boldsymbol{h}_{con} = \boldsymbol{h}_{[\text{con}]} + \boldsymbol{h}_{[/\text{con}]} \in \mathbb{R}^d, \quad (5)$$

$$\boldsymbol{h}_{[\text{CLS}]} = \boldsymbol{h}_1 \in \mathbb{R}^d, \quad (6)$$

where $\boldsymbol{h}_i \in \mathbb{R}^d$ is the text embedding of the $i$-th token in $T$, $d$ is the dimension of embedding, $N$ is the number of tokens in $T$, and $\boldsymbol{h}_{[\text{cor}]}$, $\boldsymbol{h}_{[/\text{cor}]}$, $\boldsymbol{h}_{[\text{con}]}$ and $\boldsymbol{h}_{[/\text{con}]}$ are contextual representations of tokens [cor], [/cor], [con] and [/con] respectively.

## 4.2 Attribute Siamese Network

To model the attributes constructed by activating imagination and association based on commonsense knowledge, we leverage graph neural networks, which are representative models for capturing the imaginative and associative processes in the human brain (Bessadok et al., 2022). Given that similar attributes are scarce, we design a layer-wise refinement process to filter irrelevant information gradually and learn the most suitable similar attributes. Specifically, we propose an attribute siamese network with two attribute graph subnetworks to refine similar attributes layer by layer on candidate attributes of the source concept and target concept. In addition, the core word is the only context of source and target concepts, which

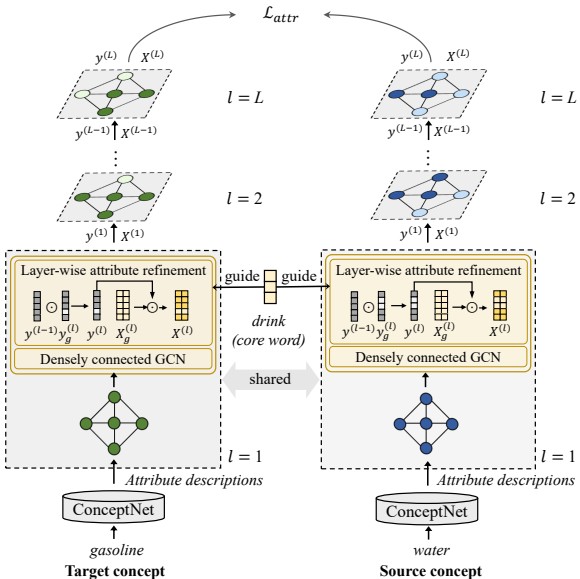

Figure 3: Overview architecture of our proposed attribute siamese network.

can provide valuable clues for inference, thus we use it as the guided information in the process of refinement. Overall architecture of the attribute siamese network is shown in Fig. 3.

### 4.2.1 Attribute Graph Construction

To obtain the candidate attributes of source and target concepts, we search a freely available commonsense knowledge resource ConceptNet (Speer et al., 2017) and acquire phrases in relations with source and target concepts as candidate attribute descriptions. For a metaphorical word pair (e.g., drink gasoline), which contains a core word $w_{cor}$ (e.g., *drink*) and a *target concept* $w_{tgt}$ (e.g., *gasoline*), we can extract a set of literal word pairs from the training set (e.g., *drink water*, *drink milk* and *drink soda*), which share the same core word (e.g., *drink*) with the aforementioned metaphorical word pair. We then randomly sample one (e.g., *drink water*) from this set of literal word pairs. The concept word (e.g., *water*) in this sampled literal word pair serves as the *source concept* $w_{src}$ in our method. We define the sequence of target concept and its candidate attribute descriptions as $\mathcal{M}^t = \{m_1^t, \ldots, m_n^t\}$, where $m_1^t = w_{tgt}$, $m_{k+1}^t$ is the $k$-th candidate attribute description extracted from ConceptNet, and the number of candidate attribute descriptions is $n - 1$. Similarly, we define the sequence of the source concept and its candidate attribute descriptions as $\mathcal{M}^s = \{m_1^s, \ldots, m_n^s\}$, where $m_1^s = w_{src}$. We convert $\mathcal{M}^t$ and $\mathcal{M}^s$ into

hidden representations $E^t = [e_1^t, \ldots, e_n^t]^\top \in \mathbb{R}^{n \times \hat{d}}$ and $E^s = [e_1^s, \ldots, e_n^s]^\top \in \mathbb{R}^{n \times \hat{d}}$ with BERT and a liner projection:

$$h_k^* = \text{BERT}([\text{CLS}] \, m_k^*)_{[\text{CLS}]}, \qquad (7)$$

$$e_k^* = W_{cor} h_k^* + b_{cor}, \qquad (8)$$

where $* \in \{t, s\}$, $k \in [1, n]$, $h_k^* \in \mathbb{R}^d$ denotes the representation of [CLS] embedded with BERT for $m_k^*$, $W_{cor} \in \mathbb{R}^{\hat{d} \times d}$ and $b_{in} \in \mathbb{R}^{\hat{d}}$ are trainable parameters, and $\hat{d}$ is the dimension of node features in the graph.

We construct attribute graphs of target concept $\mathbb{G}^t$ and source concept $\mathbb{G}^s$ with $E^t$ and $E^s$ as node feature matrices respectively. We use an adjacency matrix $A^t \in \mathbb{R}^{n \times n}$ to represent the relations of nodes in $\mathbb{G}^t$, where $n$ is the number of nodes. $(A^t)_{1,j+1} = 1$ and $(A^t)_{j+1,1} = 1$ represent there exists a relation between target concept and its $j$-th candidate attribute description ($j \in [1, n-1]$), and $(A^t)_{i,j} = 1$ represents the $i$-th and the $j$-th candidate attribute descriptions have the same relation with target concept otherwise 0. We construct $A^s \in \mathbb{R}^{n \times n}$ for $\mathbb{G}^s$ in the same way.

### 4.2.2 Layer-wise Attribute Refinement

Our attribute siamese network contains two attribute graph sub-networks sharing the same architecture and parameters. Each attribute graph sub-network consists of $L$ attribute refinement layers which refine the similar attribute information layer by layer using the core word as a guided clue. To obtain a better graph representation, we employ densely connected GCN (Guo et al., 2019b,a) as the graph encoder in our attribute refinement layers, which has advantages of strengthening feature propagation, alleviating the vanishing-gradient problem and encouraging feature reuse compared to traditional GCN (Kipf and Welling, 2017). The $l$-th attribute refinement layer contains a densely connected GCN and a refinement module guided by the embedding of core word $\boldsymbol{h}_{cor}$:

$$X_g^{(l)} = \text{DenseGCN}(X^{(l-1)}, A), \qquad (9)$$

$$X^{(l)}, y^{(l)} = \text{Refine}(X_g^{(l)}, y^{(l-1)}, \boldsymbol{h}_{cor}), \qquad (10)$$

where $l \in [1, L]$ is the index of attribute refinement layer, $A \in \mathbb{R}^{n \times n}$ is the adjacency matrix, $X^{(l-1)} \in \mathbb{R}^{n \times \hat{d}}$ is the output attribute vectors of the $(l-1)$-th layer and serves as input feature matrix of the attribute graph $\mathbb{G}^{(l)}$ with $n$ nodes, $\text{DenseGCN}(\cdot)$ is the densely connected GCN, $X_g^{(l)} \in \mathbb{R}^{n \times \hat{d}}$ is

the output of $\text{DenseGCN}(\cdot)$, and $y^{(l)} \in \mathbb{R}^n$ is the attention vector in the $l$-th layer.

Specifically, $\text{Refine}(\cdot)$ is formulated as:

$$X_c^{(l)} = \mathbf{1}_n (W_{cor} \boldsymbol{h}_{cor} + b_{cor})^\top \odot X_g^{(l)}, \qquad (11)$$

$$y_g^{(l)} = \text{Cos}(X_g^{(l)}, X_c^{(l)}), \qquad (12)$$

$$y^{(l)} = \text{Softmax}(\text{Norm}(y_g^{(l)} \odot y^{(l-1)}))), \qquad (13)$$

$$X^{(l)} = \text{Norm}(X_g^{(l)} \odot (y^{(l)} \mathbf{1}_{\hat{d}}^\top)), \qquad (14)$$

where $\mathbf{1}_k$ is a vector of size $k$ with all the components being 1, $W_{cor} \in \mathbb{R}^{\hat{d} \times d}$ and $b_{in} \in \mathbb{R}^{\hat{d}}$ are trainable parameters, $\odot$ represents the element-wise matrix multiplication, $\text{Cos}(\cdot)$ calculates cosine similarity between the $k$-th row vector in $X_g^{(l)}$ and the $k$-th row vector in $X_c^{(l)}$ resulting the $k$-th value in $y_g^{(l)} \in \mathbb{R}^n$, $\text{Norm}(\cdot)$ is the layer normalization operation, and $y^{(l)} \in \mathbb{R}^n$ is the attention vector in the $l$-th layer ($y^{(0)}$ is a vector of all the ones). We incorporate the attention vector in the $(l-1)$-th layer to refine candidate similar attributes in Eq. (13) so that the final selected similar attributes have high attention values in all the layers. Using element-wise matrix product of $X_g^{(l)}$ and $y^{(l)} \mathbf{1}_{\hat{d}}^\top$ in Eq. (14), the information of trivial nodes is controlled. We obtain the final attribute matrix $H_a \in \mathbb{R}^{n \times d}$ with a linear projection on the output features of the last layer:

$$(\boldsymbol{h}_a)_i = W_{out} \boldsymbol{x}_i^{(L)} + b_{out}, \qquad (15)$$

where $(\boldsymbol{h}_a)_i \in \mathbb{R}^d$ is the $i$-th row vector of $H_a$, $\boldsymbol{x}_i^{(L)}$ is the $i$-th row vector of $X^{(L)}$, $W_{out} \in \mathbb{R}^{d \times \hat{d}}$ and $b_{out} \in \mathbb{R}^d$ are trainable parameters.

### 4.2.3 Attribute Mapping

We employ the above two attribute graph sub-networks on attribute graphs of target concept $\mathbb{G}^t$ and source concept $\mathbb{G}^s$ respectively, obtaining aggregated attribute vectors $\boldsymbol{h}_{attr}^t \in \mathbb{R}^d$ and $\boldsymbol{h}_{attr}^s \in \mathbb{R}^d$, and design an attribute mapping loss $\mathcal{L}_{attr}$ to learn a space where aggregated attribute vectors of target and source concepts are close, which are computed as

$$\boldsymbol{h}_{attr}^t = \sum_i (\boldsymbol{h}_a)_i^t (y_i^{(L)})^t, \qquad (16)$$

$$\boldsymbol{h}_{attr}^s = \sum_i (\boldsymbol{h}_a)_i^s (y_i^{(L)})^s, \qquad (17)$$

$$\mathcal{L}_{attr} = \|\boldsymbol{h}_{attr}^t - \boldsymbol{h}_{attr}^s\|_2, \qquad (18)$$

where the superscript $t$ denotes embeddings or values in the sub-network of target concept, the superscript $s$ denotes embeddings or values in the

sub-network of source concept, $(h_a)_i$ is the $i$-th row vector of attribute matrix $H_a$, $y_i^{(L)}$ is the $i$-th scalar value in $y^{(L)}$, and $\|\cdot\|_2$ denotes mean squared L2 norm.

### 4.3 Domain Contrastive Learning

According to CMT, for the same core word, the concept words in metaphorical word pairs (e.g., *gasoline* in Fig. 1) can be regarded as part of **target domain** and the concept words in literal word pairs (e.g., *soda* and *water* in Fig. 1) can be regarded as part of **source domain**. To learn an embedding space with knowledge about the disparity of concepts in source and target domains, we propose the domain contrastive learning strategy, which makes the representation of concept words similar to the concept words in the same domain (target/source domain) and dissimilar to the concept words in the other domain (source/target domain) for the word pairs with the same core word.

Given an anchor word pair $p$ (containing a core word $w_{cor}$ and a concept word $w_{con}$) and label $l$, the positive word pair $p^+$ is the same as the anchor word pair, i.e. $l^+ = l, p^+ = p$. Following the previous convention (Gao et al., 2021), we pass the $p$ ($=p^+$) to the text encoder by applying the standard dropout twice, then we can obtain two different embeddings of $p$ and $p^+$ respectively. In contrast, we sample $n^-$ negative word pairs $P^- = \{(p_i^-, l_i^-)\}_{i=1}^{n^-}$ from the training dataset, which share the same core word with the anchor word pair but have opposite labels, i.e. $l_i^- \neq l, (w_{cor}^-)_i = w_{cor}$. Using the text encoder in Section 4.1, we can get the hidden vector $h_{con}$ of anchor concept word $w_{con}$, the hidden vector $h_{con}^+$ of positive concept word $w_{con}^+$ in $p^+$ and the hidden vectors $H_{con}^- = \{(h_{con}^-)_i\}_{i=1}^{n^-}$ of negative concept words $\{(w_{con}^-)_i\}_{i=1}^{n^-}$ in $P^-$ calculated by Eqs. (3) and (5). The domain contrastive learning loss is computed as:

$$\mathcal{L}_{dcl} = -(\log \frac{e^{f(h_{con}^+, h_{con})}}{e^{f(h_{con}^+, h_{con})} + \sum_{i=1}^{n^-} e^{f((h_{con}^-)_i, h_{con})}}$$

$$+ \log \frac{e^{f(h_{con}, h_{con}^+)}}{e^{f(h_{con}, h_{con}^+)} + \sum_{i=1}^{n^-} e^{f((h_{con}^-)_i, h_{con}^+)}})/2,$$

$$(19)$$

where $f(s_1, s_2) = \frac{s_1^\top s_2}{\tau \|s_1\| \cdot \|s_2\|}$ is the cosine similarity function with temperature parameter $\tau$.

| Dataset | Type | Train | Dev | Test | Total | %M | #UC |
|---------|------|-------|-----|------|-------|------|-----|
| MOH | SVO | - | - | - | 647 | 45.40 | 215 |
| TSV | AN | 1568 | 200 | 200 | 1968 | 50.00 | 687 |
| GUT | AN | - | - | - | 8592 | 53.60 | 23 |
| VUA-WP | SVO | 3418 | 426 | 426 | 4270 | 50.00 | 646 |
| | AN | 2682 | 334 | 334 | 3350 | 50.00 | 604 |

Table 1: Statistics of datasets. **%M** denotes the percentage of metaphorical samples in the dataset and **#UC** denotes the number of unique core words. SVO denotes subject-verb/verb-object word pair and AN denotes adjective-noun word pair.

### 4.4 Classification

Finally, we feed the aggregated attribute vector of target concept $h_{attr}^t$, the representation of core word $h_{cor}$ and classification embedding $h_{[CLS]}$ into the classifier, and adopt a cross-entropy loss function to compute classification loss $\mathcal{L}_{class}$:

$$\hat{l} = \text{softmax}(\text{MLP}(h_{attr}^t \oplus h_{cor} \oplus h_{[CLS]})), \quad (20)$$

$$\mathcal{L}_{class} = -(l)^\top \log \hat{l} \quad (21)$$

where $\oplus$ is the concatenation operation, $\text{MLP}(\cdot)$ is a two-layer multilayer perceptron with hidden dimension $\tilde{d}$, $\hat{l} \in \mathbb{R}^2$ is the predicted probability for all the labels, and $l \in \mathbb{R}^2$ is the ground truth.

### 4.5 Optimization

We optimize our method with classification loss $\mathcal{L}_{class}$, attribute mapping loss $\mathcal{L}_{attr}$ and domain contrastive learning loss $\mathcal{L}_{dcl}$:

$$\mathcal{L} = \mathcal{L}_{class} + \lambda_a \mathcal{L}_{attr} + \lambda_c \mathcal{L}_{dcl}, \quad (22)$$

where $\lambda_a$ and $\lambda_c$ are hyper-parameters.

## 5 Experiments

### 5.1 Datasets

We experiment on three publicly available word-pair metaphor datasets, and construct a new dataset **VUA-WP** based on VUA20 (Leong et al., 2020), which is the largest benchmark dataset for token-level metaphor detection task. The process of constructing the dataset VUA-WP is illustrated in Appendix A. The three benchmark datasets are as follows: (1) **MOH** (Shutova et al., 2016) contains verbal word pairs. We conduct 10-fold cross-validation for evaluation following the previous convention (Ge et al., 2022); (2) **TSV** (Tsvetkov et al., 2014) is a balanced adjective-noun word pair dataset, which contains 1768 word pairs for training and 200 word pairs for testing. We randomly

sample 200 word pairs from the original training set as the development set following the previous convention (Ge et al., 2022); (3) **GUT** (Gutiérrez et al., 2016) is an adjective-noun word-pair metaphor dataset with only 23 unique adjectives. As an adjective has more different noun associations and 98.4% word pairs in GUT never appear in TSV, we use it for testing out-of-domain generalization of models trained on TSV. Table 1 shows the statistics of these datasets.

## 5.2 Baseline Methods

We compare our method with several representative methods for word-pair metaphor detection:

- **Multimodal** (Shutova et al., 2016) combines visual and linguistic features for prediction;

- **SSN-SG** (Rei et al., 2017) is the first deep learning method for this task;

- **Concreteness** (Su et al., 2020) processes concrete word pairs with the multimodal model and processes abstract word pairs with the unimodal model;

- **EMI** (Ge et al., 2022) is the SOTA method, which models source and target concepts to benefit metaphor detection via statistic learning and a reward mechanism.

We also use two methods for text classification as baselines:

- **TextRCNN** (Lai et al., 2015) is a recurrent convolutional neural network;

- **BERT** (Devlin et al., 2019) is a pre-trained language model.

We further establish three strong baselines adapted from the representative methods in token-level metaphor detection, by replacing the target word and its context in token-level methods with the core word and the concept word in our method respectively. The details of these methods are as follows:

- **MelBERT** (Choi et al., 2021) proposes a model based on the pretrained language model inspired by metaphor identification theories MIP and SPV;

- **MrBERT** (Song et al., 2021) regards metaphor detection as a relation classification task, which cares about a kind of relations between a target word and its context inspired by MIP;

- **MisNet** (Zhang and Liu, 2022) is the SOTA method for token-level metaphor detection, which incorporates MIP and SPV into their linguistics enhanced network.[1]

We also use ChatGPT with advanced prompting strategies as baselines. Details about experiments on ChatGPT are shown in Appendix D.

## 5.3 Implementation Details

We use accuracy and macro-average F1 for evaluation. The knowledge in ConceptNet is organized in $\{s, r, o\}$ tuple format, where $s$ is the phrase subject of the tuple, $r$ is the relation of the tuple, and $o$ is the phrase object of the tuple. We regard the concept word as the phrase subject and search phrase objects which have relations with the phrase subject as candidate attribute descriptions to construct attribute graphs in attribute siamese network. In the segment embedding of the text encoder, 0, 1, 2, 3 denote the special token, the core word and its basic definition, the concept word and its basic definition, and the part-of-speech token respectively. We report the mean and standard deviation of 5 runs with different random seeds in our experiments. Other details are illustrated in Appendix B.

## 5.4 Main Results

From the experimental results shown in Table 2, we can see that our proposed method outperforms all the baselines, which verifies the effectiveness of modeling attribute likeness and domain inconsistency simultaneously for metaphor detection. Generic methods for text classification (TextRCNN and BERT) perform poorly on metaphor detection which needs high-level semantic understanding. MelBERT, MrBERT and MisNet use special network designs on pre-trained models inspired by linguistic theories (SPV and MIP), achieving comparable results with the previous state-of-the-art method EMI. However, SPV and MIP only capture contextual clues of metaphor, neglecting the implicit information of attributes and domains conveyed by metaphors. This is the reason why these methods perform worse than EMI and our method,

---

[1]For a fair comparison, we replace marker tokens in Mr-BERT with ours. We replace the basic usage of words and POS tags in MisNet with our word definitions and POS tags.

| Method | MOH | | TSV | | VUA-WP | |
|---|---|---|---|---|---|---|
| | F1 | Acc. | F1 | Acc. | F1 | Acc. |
| TextRCNN (Lai et al., 2015) | 68.94 ± 0.82 | 69.20 ± 0.86 | 72.25 ± 1.33 | 73.63 ± 1.16 | 60.95 ± 1.45 | 64.08 ± 1.13 |
| BERT[†] (Devlin et al., 2019) | 70.57 ± 1.63 | 71.22 ± 1.59 | 81.70 ± 0.59 | 81.90 ± 0.58 | 74.29 ± 1.09 | 74.42 ± 0.94 |
| MelBERT[†] (Choi et al., 2021) | 74.63 ± 1.13 | 74.98 ± 1.14 | 86.46 ± 1.31 | 86.50 ± 1.30 | 75.63 ± 0.80 | 75.71 ± 0.81 |
| MrBERT[†] (Song et al., 2021) | 74.03 ± 0.52 | 74.35 ± 0.50 | 80.84 ± 2.02 | 81.00 ± 2.02 | *76.02 ± 1.37* | *76.05 ± 1.37* |
| MisNet[†] (Zhang and Liu, 2022) | 75.09 ± 0.56 | 75.61 ± 0.58 | 84.39 ± 2.52 | 84.50 ± 2.43 | 75.72 ± 0.97 | 75.79 ± 0.97 |
| Multimodal (Shutova et al., 2016) | 75.00 | - | 79.00 | - | - | - |
| SSN-SG (Rei et al., 2017) | 74.20 | 74.08 | 80.10 | 82.20 | - | - |
| Concreteness (Su et al., 2020) | 68.00 | 64.00 | 85.00 | 84.00 | - | - |
| EMI[★] (Ge et al., 2022) | *75.60* | *75.90* | *86.60* | *87.00* | - | - |
| Ours (BERT)[†] | **79.52 ± 0.55** | **79.83 ± 0.60** | 89.98 ± 1.53 | 90.00 ± 1.52 | **76.40 ± 0.11** | **76.47 ± 0.10** |
| Ours (RoBERTa)[★] | 79.35 ± 0.45 | 79.74 ± 0.41 | **90.48 ± 0.84** | **90.80 ± 0.77** | 75.67 ± 0.41 | 75.74 ± 0.43 |

Table 2: Comparison between our method and baselines. The best results are in bold font and the best results of baselines are underlined. † and ★ denote the method using BERT and RoBERTa as text encoder respectively.

| Ablation | MOH | | TSV | | VUA-WP | |
|---|---|---|---|---|---|---|
| | F1 | Acc. | F1 | Acc. | F1 | Acc. |
| Ours (BERT) | **79.52 ± 0.55** | **79.83 ± 0.60** | **89.98 ± 1.53** | **90.00 ± 1.52** | **76.40 ± 0.11** | **76.47 ± 0.10** |
| w/o $\mathcal{L}_{dcl}$ | 78.90 ± 0.80 | 79.19 ± 0.84 | 85.78 ± 1.26 | 85.83 ± 1.25 | 72.78 ± 0.92 | 72.95 ± 1.03 |
| w/o $\mathcal{L}_{attr}$ | 79.47 ± 0.39 | 79.78 ± 0.37 | 85.27 ± 1.03 | 85.33 ± 1.03 | 74.01 ± 1.55 | 74.11 ± 1.53 |
| w/o *attribute siamese network* | 79.30 ± 0.51 | 79.62 ± 0.43 | 84.04 ± 0.82 | 84.10 ± 0.86 | 74.23 ± 1.43 | 74.53 ± 1.26 |
| w/o *core-word guidance* | 79.07 ± 0.80 | 79.47 ± 0.80 | 86.06 ± 1.69 | 86.10 ± 1.71 | 74.31 ± 0.81 | 74.50 ± 0.64 |
| w/o *DenseGCN* | 78.90 ± 0.51 | 79.22 ± 0.62 | 87.39 ± 1.79 | 87.40 ± 1.80 | 75.00 ± 0.74 | 75.10 ± 0.72 |
| w/o *part-of-speech tag* | 79.41 ± 0.51 | 79.71 ± 0.46 | 88.64 ± 0.88 | 88.67 ± 0.85 | 75.50 ± 0.31 | 75.50 ± 0.32 |
| w/o *marker tokens* | 78.88 ± 0.66 | 79.19 ± 0.68 | 88.63 ± 0.82 | 88.67 ± 0.85 | 75.51 ± 1.30 | 75.58 ± 1.29 |
| w/o *segment ID* | 79.18 ± 0.73 | 79.53 ± 0.75 | 84.66 ± 1.18 | 84.70 ± 1.16 | 74.47 ± 0.10 | 74.55 ± 0.92 |
| w/o *definition* | 77.80 ± 0.10 | 78.10 ± 0.13 | 85.34 ± 2.64 | 85.40 ± 2.61 | 74.62 ± 2.10 | 74.89 ± 1.71 |

Table 3: Results of ablation study. The best results are in bold font.

which indicates CMT is more explainable than both SPV and MIP. Although the previous SOTA method EMI considers the incongruity of source and target concepts, its performances fall far behind ours, which shows the similar attribute information they ignore is important for metaphor detection.

## 5.5 Ablation Study

We conduct the ablation study to evaluate the impact of components in our method, using the following variants: (a) **w/o** $\mathcal{L}_{dcl}$ removes the domain contrastive learning strategy in our method; (b) **w/o** $\mathcal{L}_{attr}$ removes the attribute mapping loss from final training objective; (c) **w/o *attribute siamese network*** slashes the whole attribute siamese network from our method; (d) **w/o *core-word guidance*** uses the attribute siamese network without core word as guidance to replace the attribute siamese network in our method; (e) **w/o *DenseGCN*** replaces the DenseGCN in our method with traditional GCN; (f) **w/o *part-of-speech tag*** removes the part-of-speech information in the input; (g) **w/o *marker tokens*** drops the special markers ([cor], [/cor], [con] and

[/con]) and uses the embedding of first and last tokens in core/concept word to replace the embedding of [cor]/[con] and [/cor]/[/con] respectively; (h) **w/o *segment ID*** uses 0 and 1 to represent special token ([CLS] and [SEP]) and other tokens in segment embeddings; (i) **w/o *definition*** ablates the basic definitions of core and concept words in the input.

Experimental results in Table 3 show that removing the domain contrastive learning strategy reduces the performances, which verifies the effectiveness of domain-inconsistency information for metaphor detection. To evaluate the effectiveness of attribute siamese network, we first directly ablate attribute mapping loss from our method, resulting in significant performance declines. Then we directly remove the attribute siamese network, the performances also drop consistently. Above two variants show the effectiveness of attribute information for metaphor detection. The removal of core-word guidance strategy in attribute siamese network reduces the performances, which verifies that the core word can guide our method to

| Method | F1 | Acc. |
|---|---|---|
| MelBERT (Choi et al., 2021) | 81.72 ± 0.75 | 81.74 ± 0.74 |
| MrBERT (Song et al., 2021) | 76.68 ± 0.88 | 76.71 ± 0.88 |
| MisNet (Zhang and Liu, 2022) | 81.03 ± 1.44 | 81.06 ± 1.43 |
| Ours (BERT) | **86.32 ± 1.29** | **86.35 ± 1.31** |

Table 4: Comparison between our method and baselines trained on TSV training set and tested on GUT.

learn similar attributes between source and target concepts for metaphor detection. When replacing the DenseGCN with traditional GCN, the performances on all three datasets decrease, thus verifying the advantage of DenseGCN. In addition, when removing the tricks in our method, including the part-of-speech tag, marker tokens and segment ID, models perform worse, as these tricks are important for our method to distinguish information about the core word and the concept word. When definitions are aborted, our method tends to overfit on relatively small word-pair datasets, leading to poorer performances. Besides, the performance drops of all the ablation experiments on MOH, especially when removing the attribute mapping loss, are relatively slight. This phenomenon could be attributed to the small dataset size of MOH, which only has 647 samples. However, the ablation experiments conducted on the other two larger datasets result in significant performance drops, verifying the effectiveness of the different components in our method.

### 5.6 Generalization Ability Analysis

To evaluate the generalization ability of our method, we use models trained on TSV training set to test on GUT. The experimental results in Table 4 show that our method outperforms all the comparative methods, thus verifying that our method shows promising ability of out-of-domain generalization to handle unseen concepts. Our method mines the implicit information of attributes and domains, which is more explainable and effective than comparative methods using surface-level clues of metaphor.

### 5.7 Hyper-parameter Analysis

**Hyper-parameters for Losses** We tune the hyper-parameters for attribute mapping loss $\lambda_a$ and domain contrastive learning loss $\lambda_c$ using grid search (i.e. [0.0001, 0.001, 0.01, 0.1, 1, 10, 100]) on three datasets. Experimental results in Fig. 4 show that our method achieves the best performances on TSV and VUA-WP when $\lambda_a = 0.001$ and $\lambda_c = 0.001$, and the performances drop sharply when $\lambda_a$ and $\lambda_c$ are too large. For MOH,

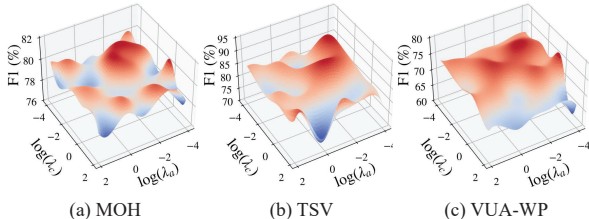

| (a) MOH | (b) TSV | (c) VUA-WP |
|---|---|---|

Figure 4: Results of our proposed method with different hyper-parameters for losses.

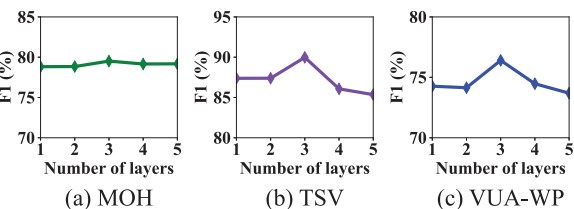

| (a) MOH | (b) TSV | (c) VUA-WP |
|---|---|---|

Figure 5: Results of our method with different number of attribute refinement layers.

our method achieves the best performance when $\lambda_a = 0.01$ and $\lambda_c = 0.001$. MOH is relatively small than other datasets leading to unstable performances with different hyper-parameters.

**Number of Attribute Refinement Layers** To analyze the impact of the number of attribute refinement layers in attribute siamese network, we experiment on varying the number of attribute refinement layers from 1 to 5. The results are shown in Fig. 5. Models on three datasets all achieve best performances with three attribute refinement layers, indicating that too shallow network is unable to capture attribute information while too deep network with plentiful parameters may lead to overfit.

## 6 Conclusion

In this paper, we propose an attribute likeness and domain inconsistency learning framework for word-pair metaphor detection. Inspired by the conceptual metaphor theory, our framework models attribute likeness with an attribute siamese network via layer-wise attribute refinement graphs and learns domain inconsistency with a domain contrastive learning strategy. Experimental results show that our method significantly outperforms the previous word-pair and token-level SOTA methods, verifying the effectiveness of our proposed framework for metaphor detection.

## Limitations

Our work has some limitations. Firstly, the generalization of our method on other types of word pairs needs to be further explored. Verb-noun and adjective-noun word pairs are the most frequently used and basic linguistic forms in metaphorical expressions. Current studies focus on above two types of word pairs, and our work follows this convention. In piratical situations, metaphor also occurs in verb-adverb word pairs, which needs further study in the future. In addition, we use ConceptNet as the external knowledge resource to obtain candidate attributes, whose types of relations are limited. The effectiveness of our method using other external knowledge resources can be further explored.

## Acknowledgements

This work is supported in part by the National Natural Science Foundation of China under Grants #72293575, #62206287 and #72225011. We thank the anonymous reviewers for the valuable comments.

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

| Type | DEP | gov. or dep. | | POS | |
|---|---|---|---|---|---|
| | | $w_{cor}$ | $w_{con}$ | $w_{cor}$ | $w_{con}$ |
| SVO | dobj | gov. | dep. | v. | n. |
| | nsubjpass | gov. | dep. | v. | n. |
| | xsubj | gov. | dep. | v. | n. |
| | agent | gov. | dep. | v. | n. |
| AN | amod | dep. | gov. | adj. | n. |
| | nsubj | gov. | dep. | adj. | n. |

Table 5: The part-of-speech and dependency information of subject-verb/verb-object (SVO) and adjective-noun (AN) candidate word pairs. *DEP* is short for *stanford typed dependencies*, *POS* is short for *part-of-speech*, *gov.* is short for *governor*, *dep.* is short for *dependent*, $w_{cor}$ is the core word and $w_{con}$ is the concept word.

| Relation | Description |
|---|---|
| HasProperty | A has B as a property |
| SymbolOf | A symbolically represents B |
| CausesDesire | A makes someone want B |
| RelatedTo | A is related to B |
| IsA | A is a subtype or a specific instance of B |
| PartOf | A is a part of B |
| HasA | B belongs to A |
| UsedFor | A is used for B |
| CapableOf | Something that A can typically do is B |
| AtLocation | A is a typical location for B |
| Causes | A and B are events, and it is typical for A to cause B |
| HasSubevent | A and B are events, and B happens as a subevent of A |
| HasFirstSubevent | A is an event that begins with subevent B |
| HasLastSubevent | A is an event that concludes with subevent B |
| HasPrerequisite | In order for A to happen, B needs to happen |
| ReceivesAction | B can be done to A |
| MadeOf | A is made of B |

Table 6: Relations in ConceptNet we used for our method. *A* denotes the phrase subject and *B* denotes the phrase object.

## A  Dataset Construction

We construct a word-pair metaphor dataset VUA-WP from VUA20 (Leong et al., 2020), which is the largest benchmark dataset for token-level metaphor detection task. Each word in sentences of VUA20 is labelled as metaphorical or literal. We use Stanford CoreNLP[1] to extract all the dependency relations and collect all the candidate word pairs with possible dependency relations according to Table 5. After that, we label these word pairs with four rules:

- If all the words in a subject-verb/verb-object word pair are metaphorical in VUA20, we label this word pair as metaphorical in VUA-WP;

- If all the words in a subject-verb/verb-object word pair are literal in VUA20, we label this word pair as literal in VUA-WP;

- If the adjective in an adjective-noun word pair is metaphorical in VUA20, we label this word pair as metaphorical in VUA-WP;

- If all the words in an adjective-noun word pair are literal in VUA20, we label this word pair as literal in VUA-WP.

The number of literal word pairs is far more than the number of metaphorical word pairs. We randomly sample literal word pairs to get a balanced dataset VUA-WP. After that, we randomly divide all the word pairs into training, development and test sets with a ratio of 8:1:1.

## B  Implementation Details

We use the Stanford CoreNLP to get part-of-speech information of word pairs. The basic definition of a word with its POS tag in a given word pair is extracted from the dictionary Vocabulary[2], using the first definition of this word under the same POS tag. To construct attribute graphs, we use Concept-Net to extract the attribute descriptions of concept words. If ConceptNet doesn't contain a source or target concept, we use COMET (Bosselut et al., 2019), a generative language model fine-tuned on ConceptNet (Speer et al., 2017), as a substitute to get attribute descriptions. The relations in Concept-Net we use are shown in Table 6. We get data from ConceptNet using the REST API[3] and we only search English terms in ConceptNet. We use bert-base-uncased[4] or roberta-base[5] as the text encoder. We implement our method based on Pytorch[6]. We use gradient clipping to avoid exploding gradients and the max norm of the gradients is 1. We use AdamW[7] as our optimizer and the coefficient of weight decay is 0.01. We train our models for 10 epochs with the learning rate of text encoder as $lr_{encoder}$ and the learning rate of other components as $lr_{exc.\ encoder}$. We use Dropout (Srivastava et al., 2014) to prevent overfitting and the dropout rate is 0.5. We train all our methods on one NVIDIA GeForce RTX 3090 GPU. For each dataset, the model giving best performance of macro-average F1-score in the development set is used for test

---

[1]https://stanfordnlp.github.io/CoreNLP/

[2]https://www.vocabulary.com/
[3]https://github.com/commonsense/conceptnet5/wiki/API
[4]https://huggingface.co/bert-base-uncased
[5]https://huggingface.co/roberta-base
[6]https://pytorch.org/
[7]https://pytorch.org/docs/stable/generated/torch.optim.AdamW.html

| Notation | Value | | | Description |
|---|---|---|---|---|
| | MOH | TSV | VUA-WP | |
| $N$ | 140 | 140 | 140 | maximum length of text tokens |
| $lr_{exc.\ encoder}$ | $5e^{-4}$ | $5e^{-4}$ | $5e^{-4}$ | learning rate of components except the text encoder |
| $lr_{encoder}$ | $5e^{-5}$ | $5e^{-5}$ | $5e^{-5}$ | learning rate of the text encoder |
| $L$ | 3 | 3 | 3 | number of layers of attribute siamese network |
| $n^-$ | 5 | 5 | 5 | number of negative samples in domain constrastive learning |
| $d$ | 768 | 768 | 768 | dimension of text embedding |
| $\tilde{d}$ | 768 | 768 | 768 | hidden dimension of MLP |
| $\hat{d}$ | 300 | 300 | 300 | dimension of node features in layer-wise refinement graph |
| $n$ | 100 | 100 | 100 | number of maximum nodes in layer-wise refinement graph |
| $n_{sub}$ | 2 | 2 | 2 | number of sublayers in densely connected GCN layer |
| $\tau$ | 0.05 | 0.05 | 0.05 | temperature of cosine similarity function |
| $\lambda_a$ | 0.01 | 0.001 | 0.001 | hyper-parameter for attribute mapping loss |
| $\lambda_c$ | 0.001 | 0.001 | 0.001 | hyper-parameter for domain contrastive learning loss |
| $Time_{tr}$ | about 40 minutes | about 30 minutes | about 45 minutes | training time |

Table 7: Hyper-parameter values in our proposed method.

| Word pair | Target concept | Relation | Attribute |
|---|---|---|---|
| fresh thinking | thinking | Causes (causes) | *a new thought* |
| | thinking | UsedFor (is used for) | AI |
| | thinking | Causes (causes) | *new perspectives* |
| bright candidate | candidate | RelatedTo (is related to) | *hopeful* |
| | candidate | IsA (is a type of) | person |
| | candidate | RelatedTo (is related to) | Clinton |
| strong effort | effort | RelatedTo (is related to) | *force* |
| | effort | RelatedTo (is related to) | *trying hard* |
| | effort | RelatedTo (is related to) | *endeavour* |

Table 8: Cases of metaphorical word pairs and corresponding attributes with the top three attention values in the last layer of our attribute siamese network.

set. We run experiments with extensive hyper-parameter search and provide details of the best model parameters in Table 7. We experiment with $L$ of $\{1, 2, 3, 4, 5\}$, $\lambda_a$ of $\{0.0001, 0.001, 0.01, 0.1, 1, 10, 100\}$, and $\lambda_c$ of $\{0.0001, 0.001, 0.01, 0.1, 1, 10, 100\}$.

## C Case Study

To show how the attribute siamese network captures similar attributes of source and target concepts, we provide several instances and their attributes with the top three attention scores in the final layer of our attribute siamese network in Table 8. We can see that our attribute siamese network can capture plausible attributes of concept words which assist in understanding metaphors. Considering the attributes *a new thought* and *new perspectives*, we can infer that *fresh thinking* in Table 8 implies new thoughts or perspectives. Considering the attribute *hopeful*, we can infer that *bright candidate* in Table 8 implies a candidate who may have a hopeful future. Considering the attributes

*force*, *trying hard* and *endeavour*, we can infer that *strong effort* in Table 8 implies putting great effort into something or trying hard on something.

## D Experiments on ChatGPT

**Implementation Details** We used OpenAI API (gpt-3.5-turbo)[1] for testing. A good prompt is essential for generative large language models. We explore different prompts with some advanced prompting strategies, including the standard zero-shot prompting, the zero-shot chain-of-thought (CoT) prompting (Kojima et al., 2022) and the standard few-shot prompting (Brown et al., 2020). For answer cleansing, we pick up the first "yes" or "no" encountered in the answer given by ChatGPT after converting all the uppercase characters in the answer string into lowercase characters.

**Prompt Design** Table 10 summarizes a list of template prompts used for the experiments with different prompting strategies on ChatGPT.

---

[1] https://platform.openai.com/docs/models/gpt-3-5

| Scientific artifact | License |
|---|---|
| roberta-base | MIT license |
| bert-base-uncased | Apache-2.0 |
| COMET | Apache-2.0 |
| gpt-3.5-turbo | API license |
| MOH | Unspecified |
| TSV | ODbL-1.0 license |
| GUT | Unspecified |
| VUA20 | CC BY-SA 3.0 |
| ConceptNet | CC BY-SA 4.0 |
| VUA-WP | MIT license |
| Stanford CoreNLP | GNU General Public License (v2 or later) |

Table 9: Licenses of the scientific artifacts in this paper.

**Experimental Results** Table 11 shows the comparison between our method and ChatGPT using different prompt strategies on our datasets. Although GhatGPT baselines use multiple advanced prompting strategies, they struggle on this task and their performances are far behind ours. In addition, the performances of ChatGPT on word-pair metaphor detection are sensitive to the design of prompts.

**Failure Analysis** To further explore why Chat-GPT performs poorly on word-pair metaphor detection, we observed the failure cases in our experiments with ChatGPT using zero-shot prompting on TSV dataset. Our observation shows that these cases can be roughly classified into four groups: wrong conjecture, uncertain judgment, contextual ambiguity, and common collocation. Table 12 gives some failure cases in each group. In the first group, ChatGPT makes wrong conjectures about the answers to the questions with unreasonable explanations. In the second group, ChatGPT makes uncertain judgments by stating metaphorical and literal word pairs. In the third group, ChatGPT as a general language model is also liable to make mistakes when the contextual information of word pairs is unavailable. Besides, in the fourth group, as some metaphorical word pairs such as "dirty word" and "heavy tax" are commonly used in everyday language, ChatGPT often misinterprets them as literal expressions. Wrong conjecture is the major reason for the failure in predicting the metaphorical tendencies with ChatGPT, which accounts for 55.55% of all the failure cases. Common collocation, uncertain judgment and contextual ambiguity comprise 17.46%, 14.29% and 12.70% respectively. In general, ChatGPT is often good at generating

fluent answers but sometimes may make incorrect or vague judgments on word-pair metaphor identification.

## E Licenses of Scientific Artifacts

The licenses of the scientific artifacts we used are shown in Table 9.

| Prompting strategy | No. | Template prompt |
|---|---|---|
| Zero-shot | 1 | Does the word pair "[word pair]" express metaphorical meaning? |
| | 2 | Does the word pair "[word pair]" express metaphorical meaning? Answer me with "yes" or "no". |
| | 3 | Does the word pair "[word pair]" express metaphorical meaning? Give me an answer selected from "yes" or "no". |
| | 4 | Is the word pair "[word pair]" a metaphorical expression? |
| | 5 | Is the word pair "[word pair]" a metaphorical expression? Answer me with "yes" or "no". |
| | 6 | Is the word pair "[word pair]" a metaphorical expression? Give me an answer selected from "yes" or "no". |
| | 7 | Does the word pair "[word pair]" use metaphor? |
| | 8 | Does the word pair "[word pair]" use metaphor? Answer me with "yes" or "no". |
| | 9 | Does the word pair "[word pair]" use metaphor? Give me an answer selected from "yes" or "no". |
| | 10 | Given the word pair "[word pair]", determine if this word pair is a metaphorical expression. |
| | 11 | Given the word pair "[word pair]", determine if this word pair is a metaphorical expression. Answer me with "yes" or "no". |
| | 12 | Given the word pair "[word pair]", determine if this word pair is a metaphorical expression. Give me an answer selected from "yes" or "no". |
| Zero-shot CoT | 1 | Does the word pair "[word pair]" express metaphorical meaning? Answer me with "yes" or "no" step by step. |
| | 2 | Is the word pair "[word pair]" a metaphorical expression? Answer me with "yes" or "no" step by step. |
| | 3 | Does the word pair "[word pair]" use metaphor? Answer me with "yes" or "no" step by step. |
| | 4 | Given the word pair "[word pair]", determine if this word pair is a metaphorical expression. Answer me with "yes" or "no" step by step. |
| 1-shot | 1 | Question: Does the word pair "[metaphorical word pair]" express metaphorical meaning? Answer: yes. Question: Does the word pair "[literal word pair]" express metaphorical meaning? Answer: no. Question: Does the word pair "[word pair]" express metaphorical meaning? Answer: |
| | 2 | Question: Does the word pair "[metaphorical word pair]" express metaphorical meaning? Answer me with "yes" or "no". Answer: yes. Question: Does the word pair "[literal word pair]" express metaphorical meaning? Answer me with "yes" or "no". Answer: no. Question: Does the word pair "[word pair]" express metaphorical meaning? Answer me with "yes" or "no". Answer: |
| | 3 | Question: Does the word pair "[metaphorical word pair]" express metaphorical meaning? Give me an answer selected from "yes" or "no". Answer: yes. Question: Does the word pair "[literal word pair]" express metaphorical meaning? Give me an answer selected from "yes" or "no". Answer: no. Question: Does the word pair "[word pair]" express metaphorical meaning? Give me an answer selected from "yes" or "no". Answer: |
| | 4 | Question: Is the word pair "[metaphorical word pair]" a metaphorical expression? Answer: yes. Question: Is the word pair "[literal word pair]" a metaphorical expression? Answer: no. Question: Is the word pair "[word pair]" a metaphorical expression? Answer: |
| | 5 | Question: Is the word pair "[metaphorical word pair]" a metaphorical expression? Answer me with "yes" or "no". Answer: yes. Question: Is the word pair "[literal word pair]" a metaphorical expression? Answer me with "yes" or "no". Answer: no. Question: Is the word pair "[word pair]" a metaphorical expression? Answer me with "yes" or "no". Answer: |
| | 6 | Question: Is the word pair "[metaphorical word pair]" a metaphorical expression? Give me an answer selected from "yes" or "no". Answer: yes. Question: Is the word pair "[literal word pair]" a metaphorical expression? Give me an answer selected from "yes" or "no". Answer: no. Question: Is the word pair "[word pair]" a metaphorical expression? Give me an answer selected from "yes" or "no". Answer: |
| | 7 | Question: Does the word pair "[metaphorical word pair]" use metaphor? Answer: yes. Question: Does the word pair "[literal word pair]" use metaphor? Answer: no. Question: Does the word pair "[word pair]" use metaphor? Answer: |
| | 8 | Question: Does the word pair "[metaphorical word pair]" use metaphor? Answer me with "yes" or "no". Answer: yes. Question: Does the word pair "[literal word pair]" use metaphor? Answer me with "yes" or "no". Answer: no. Question: Does the word pair "[word pair]" use metaphor? Answer me with "yes" or "no". Answer: |
| | 9 | Question: Does the word pair "[metaphorical word pair]" use metaphor? Give me an answer selected from "yes" or "no". Answer: yes. Question: Does the word pair "[literal word pair]" use metaphor? Give me an answer selected from "yes" or "no". Answer: no. Question: Does the word pair "[word pair]" use metaphor? Give me an answer selected from "yes" or "no". Answer: |
| | 10 | Question: Given the word pair "[metaphorical word pair]", determine if this word pair is a metaphorical expression. Answer: yes. Question: Given the word pair "[literal word pair]", determine if this word pair is a metaphorical expression. Answer: no. Question: Given the word pair "[word pair]", determine if this word pair is a metaphorical expression. Answer: |
| | 11 | Question: Given the word pair "[metaphorical word pair]", determine if this word pair is a metaphorical expression. Answer me with "yes" or "no". Answer: yes. Question: Given the word pair "[literal word pair]", determine if this word pair is a metaphorical expression. Answer me with "yes" or "no". Answer: no. Question: Given the word pair "[word pair]", determine if this word pair is a metaphorical expression. Answer me with "yes" or "no". Answer: |
| | 12 | Question: Given the word pair "[metaphorical word pair]", determine if this word pair is a metaphorical expression. Give me an answer selected from "yes" or "no". Answer: yes. Question: Given the word pair "[literal word pair]", determine if this word pair is a metaphorical expression. Give me an answer selected from "yes" or "no". Answer: no. Question: Given the word pair "[word pair]", determine if this word pair is a metaphorical expression. Give me an answer selected from "yes" or "no". Answer: |

Table 10: Prompt design based on different prompting strategies in ChatGPT experimentation. *[word pair]* denotes the input slot of the word pair that needs to be tested. *[metaphorical word pair]* denotes the input slot of the metaphorical word pair which is randomly sampled from the training dataset. *[literal word pair]* denotes the input slot of the literal word pair which is randomly sampled from the training dataset.

| Method | No. of template prompt | MOH | | TSV | | VUA-WP | |
|---|---|---|---|---|---|---|---|
| | | F1 | Acc. | F1 | Acc. | F1 | Acc. |
| ChatGPT Zero-shot | 1 | 51.89 | 59.66 | 65.72 | 68.50 | 38.55 | 51.58 |
| | 2 | 62.02 | 65.22 | 73.38 | 74.50 | 44.72 | 53.68 |
| | 3 | 59.45 | 63.52 | 70.44 | 72.00 | 44.43 | 54.34 |
| | 4 | 48.91 | 57.81 | 72.22 | 73.50 | 42.76 | 53.95 |
| | 5 | 52.35 | 59.35 | 73.79 | 74.50 | 42.68 | 53.29 |
| | 6 | 50.67 | 58.58 | 71.17 | 72.50 | 41.53 | 53.03 |
| | 7 | 55.72 | 60.43 | 60.94 | 64.00 | 40.43 | 52.63 |
| | 8 | 52.28 | 57.65 | 61.39 | 64.00 | 43.00 | 53.55 |
| | 9 | 54.05 | 59.66 | 60.17 | 64.00 | 40.60 | 52.63 |
| | 10 | 43.33 | 54.71 | 63.76 | 66.50 | 41.50 | 53.55 |
| | 11 | 51.60 | 58.89 | 66.66 | 68.50 | 44.11 | 54.08 |
| | 12 | 56.54 | 61.21 | 74.90 | 75.50 | 46.46 | 54.47 |
| ChatGPT Zero-shot CoT | 1 | 63.55 | 63.83 | 68.16 | 69.50 | 55.53 | 55.53 |
| | 2 | 56.39 | 56.41 | 72.38 | 73.00 | 53.76 | 54.34 |
| | 3 | 55.12 | 56.88 | 61.54 | 65.00 | 56.18 | 56.18 |
| | 4 | 52.79 | 53.94 | 75.49 | 75.50 | 55.49 | 57.37 |
| ChatGPT 1-shot | 1 | 60.23 | 64.91 | 69.33 | 70.50 | 42.34 | 52.50 |
| | 2 | 57.26 | 62.75 | 73.49 | 74.50 | 41.71 | 52.50 |
| | 3 | 54.90 | 61.05 | 71.03 | 72.50 | 42.44 | 53.42 |
| | 4 | 51.27 | 59.20 | 71.20 | 73.00 | 38.92 | 51.58 |
| | 5 | 53.39 | 60.74 | 68.78 | 70.50 | 40.83 | 53.03 |
| | 6 | 50.14 | 58.89 | 62.81 | 66.50 | 39.21 | 52.11 |
| | 7 | 49.76 | 58.58 | 67.38 | 69.50 | 42.11 | 53.42 |
| | 8 | 51.97 | 59.97 | 63.54 | 66.50 | 42.60 | 53.68 |
| | 9 | 45.45 | 56.26 | 61.43 | 65.50 | 37.63 | 51.32 |
| | 10 | 52.63 | 60.28 | 75.84 | 76.50 | 42.43 | 53.68 |
| | 11 | 52.86 | 60.59 | 70.13 | 72.00 | 40.58 | 52.89 |
| | 12 | 48.74 | 58.27 | 64.84 | 68.00 | 40.91 | 53.16 |
| ChatGPT 5-shot | 1 | 65.00 | 66.31 | 63.49 | 67.00 | 48.29 | 54.87 |
| | 2 | 59.80 | 62.13 | 68.62 | 70.50 | 48.82 | 55.00 |
| | 3 | 62.27 | 64.30 | 72.20 | 73.00 | 48.26 | 53.68 |
| | 4 | 63.51 | 66.00 | 57.09 | 62.50 | 46.62 | 54.34 |
| | 5 | 60.06 | 62.60 | 68.45 | 70.50 | 46.53 | 54.21 |
| | 6 | 57.31 | 60.43 | 65.87 | 68.00 | 48.10 | 54.61 |
| | 7 | 59.44 | 62.44 | 63.06 | 66.50 | 48.07 | 54.87 |
| | 8 | 60.31 | 64.14 | 67.72 | 69.50 | 49.13 | 55.13 |
| | 9 | 55.55 | 59.97 | 63.10 | 65.50 | 49.53 | 55.39 |
| | 10 | 63.54 | 65.69 | 71.03 | 72.50 | 49.05 | 54.74 |
| | 11 | 54.91 | 59.66 | 68.45 | 70.50 | 47.21 | 55.00 |
| | 12 | 55.58 | 61.05 | 65.28 | 68.00 | 43.98 | 52.76 |
| Ours (BERT) | - | **79.52** | **79.83** | 89.98 | 90.00 | **76.40** | **76.47** |
| Ours (RoBERTa) | - | 79.35 | 79.74 | **90.48** | **90.80** | 75.67 | 75.74 |

Table 11: Comparison between our method and GhatGPT. The best results are in bold font and the best result of different template prompts for every prompting strategy is underlined.

| Category | Word pair | Prompt question | Answer of ChatGPT | Result of AIDIL |
|---|---|---|---|---|
| *Wrong conjecture* | mad man | Does the word pair "mad man" express metaphorical meaning? | Yes, the word pair "mad man" can have metaphorical meaning. It could be used to describe someone who behaves irrationally or impulsively, like a "mad scientist" who takes risks in their experiments, or a "mad genius" who has unconventional ideas. (✗) | literal (✓) |
| | blunt honesty | Does the word pair "blunt honesty" express metaphorical meaning? | No, the word pair "blunt honesty" is not a metaphor. It is a description of honesty that is direct and straightforward, without any attempt to soften or sugarcoat the truth. (✗) | metaphorical (✓) |
| *Uncertain judgment* | young moon | Does the word pair "young moon" express metaphorical meaning? | It is possible for "young moon" to express metaphorical meaning, depending on the context in which it is used. For example, the phrase could be used to represent new beginnings, growth, or innocence. However, it could also simply refer to the phase of the moon when it is visible as a thin crescent in the sky. (✗) | metaphorical (✓) |
| | red alert | Does the word pair "red alert" express metaphorical meaning? | As an AI language model, I do not have personal opinions or feelings, but I can tell you that the word pair "red alert" has a literal meaning related to emergency situations, such as the highest level of warning or readiness. However, it can also be used metaphorically to refer to situations of intense urgency, danger, or crisis outside the context of emergencies. (✗) | metaphorical (✓) |
| *Contextual ambiguity* | white anger | Does the word pair "white anger" express metaphorical meaning? | As an AI language model, I do not fully understand the context or the intention behind the phrase "white anger". Without further context, the phrase does not seem to express a metaphorical meaning. However, it is important to note that the phrase could have a different connotation or meaning, depending on the context or cultural references associated with it. (✗) | metaphorical (✓) |
| | breezy disregard | Does the word pair "breezy disregard" express metaphorical meaning? | As an AI language model, I cannot definitively determine if a given phrase expresses metaphorical meaning without the context in which it is being used. However, the combination of "breezy" and "disregard" may suggest a metaphorical meaning that suggests an easy or casual dismissal of something important or serious. (✗) | metaphorical (✓) |
| *Common collocation* | dirty word | Does the word pair "dirty word" express metaphorical meaning? | No, the word pair "dirty word" is a literal expression. It is used to describe a word that is considered vulgar or indecent in society. (✗) | metaphorical (✓) |
| | heavy tax | Does the word pair "heavy tax" express metaphorical meaning? | No, the word pair "heavy tax" expresses a literal meaning, referring to a high amount of tax that needs to be paid. (✗) | metaphorical (✓) |

Table 12: Failure cases in our experiments with ChatGPT using zero-shot prompting on TSV dataset.