# OpenReview forum: "Modeling Conceptual Attribute Likeness and Domain Inconsistency for Metaphor Detection"
_EMNLP/2023/Conference — EMNLP 2023 Main_

### Official Review · Reviewer_Gckv · 2023-08-03

**Soundness:** 3

**Excitement:**

3: Ambivalent: It has merits (e.g., it reports state-of-the-art results, the idea is nice), but there are key weaknesses (e.g., it describes incremental work), and it can significantly benefit from another round of revision. However, I won't object to accepting it if my co-reviewers champion it.

**Paper Topic And Main Contributions:**

The authors follow the Conceptual Metaphor Theory and find that previous studies ignore the similarity of attributes shared by the source and target domains. To remedy this, they propose a novel method entitled AIDIL. The method involves a Siamese neural network to compute the attribute similarity between the source and the target. Moreover, they propose a domain inconsistency loss based on the contrastive learning.

**Questions For The Authors:**

Why make use of the DenseGCN as the graph encoder?


**Reasons To Accept:**

(1)	This method designs a new loss function that considers the similarity of attributes with the inspiration of the CMT.
(2)	This paper is well writing.
(3)	In the experimental part, the authors make use of the powerful chatGPT for comparison.
(4)	The experiments are comprehensive and clear. The authors do not only analyze the impact of layers in the GNN, but also verify the impact of weights of the two proposed loss functions.


**Reasons To Reject:**

(1)	As graph construction and learning is time-consuming, I think the authors should demonstrate the computational complexity of the proposed method.
(2)	There are too many blank spaces in the main experimental results.
(3)	The authors do not analyze the impact of the quality of the attribute graphs.


**Reproducibility:**

4: Could mostly reproduce the results, but there may be some variation because of sample variance or minor variations in their interpretation of the protocol or method.

**Reviewer Confidence:**

4: Quite sure. I tried to check the important points carefully. It's unlikely, though conceivable, that I missed something that should affect my ratings.

---

> ### Author Rebuttal · Authors · 2023-08-29
>
> Many thanks for your comprehensive and insightful review. Below we shall respond to them one by one.
>
>
> ***Q1. Why make use of the DenseGCN as the graph encoder?***
>
> We use the DenseGCN as the graph encoder for the reason that it has several advantages, such as strengthening feature propagation, alleviating the vanishing-gradient problem and encouraging feature reuse (Huang et al., 2017; Guo et al., 2019). The experimental results of the ablation study on DenseGCN in Table 1 show that when replacing the DenseGCN with traditional GCN, the performances on all three datasets decrease, thus verifying the advantage of DenseGCN. We shall add these results and the analysis in the revision.
>
>
> | Method                | MOH (10-fold) | MOH (10-fold)     |  TSV           |       TSV          |      VUA-WP          |    VUA-WP             |
> |-----------------------|---------------|--------------|---------------|--------------|--------------|---------------|
> |                       | **F1**            | **Acc.**         | **F1**            | **Acc.**         | **F1**           | **Acc.**          |
> | Our method (DenseGCN) | 79.52±0.55  | 79.83±0.60 | 89.98±1.53  | 90.00±1.52 | 76.40±0.11 | 76.47±0.10  |
> | Our method (GCN)      | 78.90±0.51  | 79.22±0.62 | 87.39±1.79  | 87.40±1.80 | 75.00±0.74 | 75.10±0.72  |
>
> Table 1: Results of the ablation study on DenseGCN.
>
>
>
> ***R1. As graph construction and learning is time-consuming, I think the authors should demonstrate the computational complexity of the proposed method.***
>
> **Response:** Thanks for the suggestion. We use FLOPs to evaluate the time complexity and the number of parameters to evaluate the space complexity. From the results in Table 2, we can see that the pretrained model BERT constitutes the majority of the computational complexity of our proposed method. Although we use the graph-based network as our attribute siamese network, the computational cost of this network is just 0.14G FLOPs, which is trivial compared to that of BERT. In addition, the results in Table 3 show that the computational cost of our method is comparable to that of baselines. We shall add this analysis in the revision.
>
>
> | Module                                          | FLOPs  | #Params   |
> |-------------------------------------------------|--------|-----------|
> | Attribute Siamese Network (graph-based network) | *0.14G*  | *1,615K*    |
> | BERT                                            | 11.90G | 109,487K  |
> | Our proposed method                             | 12.05G | 116,417K  |
>
> Table 2: Computational complexity of our proposed method.
>
> | Method              | FLOPs  | #Params  |
> |---------------------|--------|----------|
> | MelBERT             | 23.81G | 112M     |
> | MrBERT              | 11.90G | 110M     |
> | MisNet              | 13.61G | 114M     |
> | EMI                 | -      | 125M+    |
> | Our proposed method | 12.05G | 116M     |
>
> Table 3: Comparison of computational complexity between baselines and our proposed method. Given that EMI did not open-source their codes, we cannot get a precise FLOPs of this method. EMI uses RoBERTa as the text encoder whose number of parameters is 125M, and thus the number of parameters of EMI is more than 125M.
>
>
>
> ***R2. There are too many blank spaces in the main experimental results.***
>
>
> **Response:** Sorry for the confusion. Some baselines (Multimodal, SSN-SG, Concreteness and EMI) did not release their source codes, resulting in the absence of their experimental results on our newly constructed dataset VUA-WP. In addition, the experimental results of these baselines in our main experimental results are retrieved from their papers, which do not provide the standard deviations.
>
>
> ***R3. The authors do not analyze the impact of the quality of the attribute graphs.***
>
> **Response:** Thanks. The quality of the attribute graphs mainly depends on edges and nodes, which are fundamental elements in graphs. We conduct two ablation studies on the quality of edges and nodes. The experimental results are shown in Table 4. When we randomly assign 50% of the edges in the attribute graph, the performances decrease across datasets. This is because edges represent the relationships between nodes, which are important for node representation learning. If we randomly delete 50% of the nodes when constructing the attribute graphs, the essential knowledge required to learn the final attribute vector may be lost, thus leading to consistent performance drops on three datasets. These experimental results verify that the quality of the attribute graphs has an impact on our method.
>
>
> | Method           | MOH (10-fold) | MOH (10-fold)         | TSV        |   TSV             |     VUA-WP          |    VUA-WP                 |
> |------------|---------------|--------------|---------------|--------------|--------------|---------------|
> |                  | **F1**            | **Acc.**         | **F1**           | **Acc.**         | **F1**        | **Acc.**         |
> | Our method  | 79.52±0.55  | 79.83±0.60 | 89.98±1.53  | 90.00±1.52 | 76.40±0.11 | 76.47±0.10  |
> | 50% random edges | 78.43±0.56  | 78.75±0.53 | 82.90±1.27  | 83.00±1.22 | 73.54±0.32 | 73.72±0.31  |
> | - 50% nodes| 78.21±0.52| 78.57±0.57| 82.68±0.91  | 82.80±0.98 | 74.11±0.19 | 74.16±0.18  |
>
> Table 4: Results of the ablation study on the edges and nodes in the attribute graph.
>
> We shall add a line chart to show the trend of performance change when varying the proportions of random edges and deleted nodes in the revision.

---

### Official Review · Reviewer_kRQ4 · 2023-08-03

**Soundness:** 4

**Excitement:**

4: Strong: This paper deepens the understanding of some phenomenon or lowers the barriers to an existing research direction.

**Paper Topic And Main Contributions:**

This paper addresses the task of word-pair metaphor detection by modeling similar attributes between source and target concepts in metaphors. It is motivated by the CMT theory (Lakoff and Johnson, 2018), which states that metaphors are about the mapping of a set of attributes between the source and target concepts). Therefore, the authors propose an approach consisting of three components, each learning a subtask. The first component is a Siamese neural network that generates an attribute vector for the given metaphor by contrasting it to other literal or metaphorical word pairs. The second component learns the domain relation between the source and target concepts via contrastive learning. Finally, the third component is the classifier that takes the attribute vector and the word pair and predicts whether it is a metaphor or not. The three components are jointly learned.

In a set of experiments, the authors compare their approach to various baselines on three datasets and demonstrate state-of-the-art results on the task. The ablation study also shows that all components implemented in this approach contribute a certain boost in performance.

UPDATE: The authors clarified my question and I will up my scores

**Questions For The Authors:**

- The opening sentence in the abstract: "Metaphor detection is an important and challenging task to identify metaphorical expression in natural language processing" .. what is that supposed to mean?

**Reasons To Accept:**

- A new approach to the task with strong results

**Reasons To Reject:**

- Experiment setting is not clear:
	- The paper needs to mention the prompts used for ChatGPT. Surprisingly, ChatGPT performed very poorly on this task.

- Paper presentation is not optimal:
	- I struggled to understand the sampling process that was followed in Section 4.2.1. I encourage the authors to show an example here.

	- Details on the process of constructing the dataset VUA-WP are missing

**Reproducibility:**

2: Would be hard pressed to reproduce the results. The contribution depends on data that are simply not available outside the author's institution or consortium; not enough details are provided.

**Reviewer Confidence:**

2: Willing to defend my evaluation, but it is fairly likely that I missed some details, didn't understand some central points, or can't be sure about the novelty of the work.

---

> ### Author Rebuttal · Authors · 2023-08-29
>
> Many thanks for your valuable suggestions and comments. Below we shall respond to them one by one.
>
> ***Q1. The opening sentence in the abstract: "Metaphor detection is an important and challenging task to identify metaphorical expression in natural language processing" .. what is that supposed to mean?***
>
> **Response:** Sorry for the confusion. We have rewritten this sentence for clarity.
>
> *$~~~~$“Metaphor detection is an important and challenging task in natural language processing, which aims to identify figurative expressions in the text, in the form of word pairs or individual words within the sentence.”*
>
>
> ***R1. Experiment setting is not clear: The paper needs to mention the prompts used for ChatGPT. Surprisingly, ChatGPT performed very poorly on this task.***
>
> **Response:** Thanks for your comment. We present details of prompts for ChatGPT with different prompting strategies in Appendix D.2 of our paper, which are as follows:
>
> | Prompting Strategy | No. | Template Prompt                                                                                                                              |
> |--------------------|-----|----------------------------------------------------------------------------------------------------------------------------------------------|
> | Zero-shot          | 1   | Does the word pair "[word pair]" express metaphorical meaning?                                                                               |
> |                    | 2   | Does the word pair "[word pair]" express metaphorical meaning? Answer me with "yes" or "no".                                                 |
> |                    | ...   | ...                               |
> |                    | 12  | Given the word pair "[word pair]", determine if this word pair is a metaphorical expression. Give me an answer selected from "yes" or "no".  |
> | Zero-shot CoT      | 1   | Does the word pair "[word pair]" express metaphorical meaning? Answer me with "yes" or "no" step by step.                                    |
> |                    | ...  |...                               |
> |                    | 4   | Given the word pair "[word pair]", determine if this word pair is a metaphorical expression. Answer me with "yes" or "no" step by step.        |
> | 1-shot | 1  | Question: Does the word pair "[metaphorical word pair]" express metaphorical meaning? Answer: yes. Question: Does the word pair "[literal word pair]" express metaphorical meaning? Answer: no. Question: Does the word pair "[word pair]" express metaphorical meaning? Answer:                                                                                                                                                                                                                                          |
> |        | ...  |...                                                                                       |
> |        | 12 | Question: Given the word pair "[metaphorical word pair]", determine if this word pair is a metaphorical expression. Give me an answer selected from "yes" or "'no". Answer: yes. Question: Given the word pair "[literal word pair]", determine if this word pair is a metaphorical expression. Give me an answer selected from "yes" or "no". Answer: no. Question: Given the word pair "[word pair]", determine if this word pair is a metaphorical expression. Give me an answer selected from "yes" or "no". Answer:  |
>
> Table 11: Template prompts used for ChatGPT baselines. [word pair] denotes the input slot of the word pair that needs to be tested. [metaphorical word pair] and [literal word pair] denote the input slots of the metaphorical and literal word pairs respectively, which are randomly sampled from the training dataset.
>
>
> In Subsection 5.4, we have added a more detailed analysis on why ChatGPT performs relatively poor on this task.
>
> *$~~~~$“GhatGPT baselines perform relatively poor on this task for the reason that understanding metaphor is a difficult task that requires deep, complex, or abstract reasoning, while ChatGPT is a general language model, which can handle some forms of reasoning but still struggles with complex reasoning tasks. In addition, the training data of metaphor detection is limited and difficult to collect in contrast to other research fields like machine translation, which further hinders ChatGPT's ability to understand metaphor.”*
>
>
>
>
> ***R2. Paper presentation is not optimal: I struggled to understand the sampling process that was followed in Section 4.2.1. I encourage the authors to show an example here.***
>
> **Response:** Thanks for the suggestion. We have carefully rewritten the sampling process and added an example in Subsection 4.2.1.
>
> *$~~~~$“For a metaphorical word pair (e.g., drink gasoline), which contains a core word $w_{cor}$ (e.g., drink) and a target concept $w_{tgt}$ (e.g., gasoline), we can extract a set of literal word pairs from the training set (e.g., drink water, drink milk and drink soda), which share the same core word (e.g., drink) with the aforementioned metaphorical word pair. We then randomly sample one (e.g., drink water) from this set of literal word pairs. The concept word (e.g., water) in this sampled literal word pair serves as the source concept $w_{src}$ in our model.”*
>
> We present the following figure to illustrate the sampling process and shall add it in the revision.
> |Metaphorical word pair|Core word|Target concept|
> |:--|:--|:--|
> |drink gasoline|*drink*|gasoline|
>
>
> ↓ Sample one literal word pair from “drink water”, “drink soda”, and “drink milk”
>
> |Sampled literal word pair|Core word|Source concept|
> |:--|:--|:--|
> |drink water|*drink*|water|
>
>
>
>
> ***R3. Paper presentation is not optimal: Details on the process of constructing the dataset VUA-WP are missing***
>
> **Response:** Thanks for your comment. We present the details on the process of constructing the dataset VUA-WP in Appendix A.1 of our paper, which are as follows:
>
> $~~~~$*“We construct a word-pair metaphor dataset VUA-WP from VUA20 (Leong et al., 2020), which is the largest benchmark dataset for the token-level metaphor detection task. Each word in sentences of VUA20 is labelled as metaphorical or literal. We use the Stanford dependency parser (Chen and Manning, 2014) to extract all the dependency relations and collect all the candidate word pairs with possible dependency relations according to Table 5. After that, we label these word pairs with four rules*:
> - *If all the words in a subject-verb/verb-object word pair are metaphorical in VUA20, we label this word pair as metaphorical;*
> - *If all the words in a subject-verb/verb-object word pair are literal in VUA20, we label this word pair as literal;*
> - *If the adjective in an adjective-noun word pair is metaphorical in VUA20, we label this word pair as metaphorical;*
> - *If all the words in an adjective-noun word pair are literal in VUA20, we label this word pair as literal.*
>
> *The number of literal word pairs is far more than the number of metaphorical word pairs. We randomly sample literal word pairs to get a balanced dataset VUA-WP. After that, we randomly divide all the word pairs into training, development and test sets with the ratio of 8:1:1.”*

---

### Official Review · Reviewer_Kgcm · 2023-08-04

**Soundness:** 3

**Excitement:**

4: Strong: This paper deepens the understanding of some phenomenon or lowers the barriers to an existing research direction.

**Paper Topic And Main Contributions:**

This paper studies the problem of word-pair metaphor detection based on conceptual metaphor theory. And this paper proposes a novel attribute siamese network to mine similar attributes between target concepts and source concepts.  Besides, this paper utilizes contrastive learning to learn domain inconsistency between source and target domains.  Experimental results on three publicly available word-pair metaphor datasets show the effectiveness of the proposed method.  And this paper demonstrate the generalization ability on a new dataset.


**Questions For The Authors:**

1. As far as I know, the input context of token-level metaphor detection methods are sentences, is it appropriate to directly use token-level methods for word-pair metaphor detection？

2. In Table 2, the performance is almost unchanged after removing the attribute mapping loss on MOH. Why?

**Reasons To Accept:**

1. This paper puts forward a novel idea that utilizes attribute likeness to improve the performance of metaphor detection. Experimental results show that the proposed method outperforms the previous word-pair methods.

2. The paper is well-written and organized. The paper compares the proposed method with various types of methods and  carries out qualitative and quantitative analysis. Besides, effective ablation study shows the effects of different parts of the proposed method.

**Reasons To Reject:**

1. As far as I know, the input context of token-level metaphor detection methods are sentences, is it appropriate to directly use token-level methods for word-pair metaphor detection？

2. In Table 2, the performance is almost unchanged after removing the attribute mapping loss on MOH. Why?

3. I doubt whether word-pair metaphor detection makes sense in the filed of metaphor detection now.

**Reproducibility:**

3: Could reproduce the results with some difficulty. The settings of parameters are underspecified or subjectively determined; the training/evaluation data are not widely available.

**Reviewer Confidence:**

4: Quite sure. I tried to check the important points carefully. It's unlikely, though conceivable, that I missed something that should affect my ratings.

---

> ### Author Rebuttal · Authors · 2023-08-29
>
> Thanks very much for your encouragement and valuable comments. Below we shall respond to them one by one.
>
>
> ***Q1/R1. As far as I know, the input context of token-level metaphor detection methods are sentences, is it appropriate to directly use token-level methods for word-pair metaphor detection?***
>
> **Response:** Unfortunately, it is not appropriate to directly use token-level methods for word-pair metaphor detection. We utilized three representative methods in token-level metaphor detection as baselines and adapted them to word-pair metaphor detection. Recent studies in token-level metaphor detection tackle this task as a word classification task, which take a sentence and a target word within this sentence as input and then predict the metaphoricity of the target word. In these baseline experiments, we replaced the target word and its context in token-level methods with the core word and the concept word in our method respectively. We shall add these descriptions of token-level baselines in the revision.
>
>
> ***Q2/R2. In Table 2, the performance is almost unchanged after removing the attribute mapping loss on MOH. Why?***
>
> **Response:** The performance drops of all the ablation experiments on MOH, including removing the attribute mapping loss, are relatively slight. This phenomenon can be attributed to the small dataset size of MOH, which only has 647 samples. However, the ablation experiments conducted on the other two larger datasets result in significant performance drops, verifying the effectiveness of the attribute mapping loss. We shall include this discussion in the revision.
>
>
> ***R3. I doubt whether word-pair metaphor detection makes sense in the field of metaphor detection now.***
>
> **Response:** Thanks for pointing this out. Word-pair metaphor detection makes sense in the field of metaphor detection now for the following reasons:
> 1. Word pair is the basic and most frequently used linguistic form to express metaphor (Tsvetkov et al., 2014) and word-pair metaphor detection has consistently been an important research topic in metaphor detection (Rei et al., 2017; Su et al., 2020; Ge et al., 2022).
> 2. Word-pair metaphor detection can benefit many NLP tasks. For example, it can assist machine translation systems in achieving more accurate and refined translations (Mao et al., 2018).
> 3. The word pair, which offers a structured context, can serve as a bridge between linguistics, cognitive science and computational approaches, facilitating interdisciplinary insights into the essence of metaphors (Lakoff and Johnson, 2008; Ge et al., 2022).

---

### Meta-Review · Area_Chair_V2x4 · 2023-09-17

**Recommendation:** 4

**Metareview:**

Inspired by conceptual metaphor theory, this paper presents a new model for metaphor detection, (specifically for pairs of words in context), using a Siamese network and contrastive loss. The reviewers were impressed with the quality of writing and the quality of the experiments ("comprehensive and clear").

In terms of scores, reviewers were balanced between good and strong / ambivalent and strong. However, the authors provided extensive responses in their rebuttal, including updated language and promised changes, which largely seem to mitigate the (mostly minor) concerns. In addition, the most favorable reviewer was the most engaged with the authors during the discussion.

Although I am not convinced that the relatively poor performance of ChatGPT can be taken as conclusive (given the number of degrees of freedom in such a setup), that part is not particularly important for this paper to succeed. Rather, it can stand on its strong experimental results (relative to other methods), the quality of its writing, and its contributions to an interesting area.

---

### Decision · Program_Chairs · 2023-10-07

**Decision:**

Accept-Main

**Comment:**

Inspired by conceptual metaphor theory, this paper presents a new model for metaphor detection, (specifically for pairs of words in context), using a Siamese network and contrastive loss. The reviewers were impressed with the quality of writing and the quality of the experiments ("comprehensive and clear").

In terms of scores, reviewers were balanced between good and strong / ambivalent and strong. However, the authors provided extensive responses in their rebuttal, including updated language and promised changes, which largely seem to mitigate the (mostly minor) concerns. In addition, the most favorable reviewer was the most engaged with the authors during the discussion.

Although I am not convinced that the relatively poor performance of ChatGPT can be taken as conclusive (given the number of degrees of freedom in such a setup), that part is not particularly important for this paper to succeed. Rather, it can stand on its strong experimental results (relative to other methods), the quality of its writing, and its contributions to an interesting area.